# Surgical Outcomes for Upper Lumbar Disc Herniation: Decompression Alone versus Fusion Surgery

**DOI:** 10.3390/jcm8091435

**Published:** 2019-09-11

**Authors:** Tung-Yi Lin, Ying-Chih Wang, Chia-Wei Chang, Chak-Bor Wong, You-Hung Cheng, Tsai-Sheng Fu

**Affiliations:** Department of Orthopedic Surgery, Chang Gung Memorial Hospital, Keelung branch, Keelung 204 and School of medicine, Chang Gung University, Taoyuan 333, Taiwan

**Keywords:** upper lumbar disc herniation, spinal fusion, decompression, surgical outcomes

## Abstract

Upper lumbar herniated intervertebral disc (HIVD), defined as L1-2 and L2-3 levels, presents with a lower incidence and more unfavorable surgical outcomes than lower lumbar levels. There are very few reports onthe appropriate surgical interventions for treating upper lumbar HIVD. This study aimed to evaluate the surgical outcome of decompression alone, when compared with spinal fusion surgery. A retrospective study involving a total of 7592 patients who underwent surgery due to HIVD in our institution was conducted. A total of 49 patients were included in this study: 33 patients who underwent decompression-only surgery and 16 patients who underwent fusion surgery. Demographic data, perioperative information, and functional outcomes were recorded. The visual analog scale (VAS) scores showed improvement in both groups postoperatively. The three-month postoperative Oswestry Disability Index score was significantly better in the fusion group. Additionally, 10 patients (76.9%) in the decompression group and 5 patients (83.3%) in the fusion group reported improvement in preoperative motor weakness. The final “satisfactory” rate was 66.7% in the decompression group and 93.8% in the fusion group (*p* = 0.034). The overall surgical outcomes of patients with upper lumbar HIVD were satisfactory in this study without any major complications. More reliable satisfactory rates and better functional scores at the three-month postoperative follow-up were reported in the fusion group.

## 1. Introduction

Upper lumbar herniated intervertebral disc(HIVD), defined as L1-2 and L2-3 levels, presents with a low incidence of about 1% of all lumbar HIVD [1]. Due to the anatomical characteristics of the upper lumbar spine, the clinical symptoms and signs are non-specific and different from those in lower lumbar HIVD [2,3]. An upper lumbar HIVD, along with the anatomy of the conusmedullaris and narrow spinal canal at these levels, can cause the compression of multiple roots resulting in various radiculopathies [4,5,6].

The surgical outcome for upper lumbar HIVD showed to be more unpredictable than for lower lumbar HIVD in results from previous studies in early years [7]. Sanderson et al. [2] presented a chart review study that showed less than 60% of patients in the L1-2 and L2-3 group experienced any improvement of their pain postoperatively. Twenty percent of patients in the L1-2 and L2-3 group required a fusion procedure and this group also had significantly worse outcomes in their economic or functional status. Recently, Kim et al. [4] reported acceptable outcomes after decompression surgery by conventional laminectomy or a posterior transdural approach to perform discectomy. Preoperative symptoms improved significantly in 80.5% of patients.

Due to the characteristics of the upper lumbar levels, spine surgeons should perform neurolysis carefully and avoid violent manipulation over the dura. Limited laminotomy may be considered inadequate to explore the herniated disc and sometimes extended decompression surgery may be necessary, especially for patients with a central disc at an upper lumbar level. Indeed, iatrogenic spinal instability can develop postoperatively [8]. Lumbar fusion surgery should not be routinely performed in patients who have an absence of deformity or instability [9]. However, fusion surgery may be a reliable method to avoid a recurrence of disc herniation and postoperative spinal instability.

The rare frequency of upper lumbar HIVD results in less data and context in which to define appropriate surgical interventions. To our current knowledge, there are no reports of comparative analyses focusing on decompression alone versus fusion surgery in patients with upper lumbar HIVD. The purpose of this study is to evaluate the surgical outcomes of decompression alone compared with spinal fusion surgery in treating upper lumbar HIVD.

## 2. Materials and Methods

Between January 2013 and December 2017, 51 patients had a diagnosis of upper lumbar HIVD (L1-2 and L2-3 levels) from 7592 patients who underwent surgical intervention due to HIVD in our institution. Two patients were excluded from this comparative study because their discomfort was relieved after receiving a nerve block procedure. There were a total of 49 patients enrolled in this study: 10 patients involved the L1-2 level, 38 patients involved the L2-3 level and one patient involved both levels. Demographic data including age, sex, and muscle power status were collected. All patients received radiography and magnetic resonance imaging that were performed for establishing the diagnosis. All patients had failed conservative treatment and, subsequently, surgery was indicated.

The patients were divided into two groups depending on whether they received decompression alone or a fusion procedure. The operation performed was the standard spinal surgery after localization by fluoroscopy. During the surgery, the nerve root was visualized after decompression and mobilized gently by nerve hook retractor to expose the herniated disc fragment. Whenever doubt exists regarding identification of the nerve structure, a wide laminectomy might be necessary. The surgical goal was to achieve an acceptable decompression of nerve structures. In the patients who received fusion surgery, transpedicular pedicle screws and interbody fusion with cage were performed to avoid iatrogenic instability. The pain scores were evaluated (the most severe pain, either leg pain or back pain) by visual analogues scale (VAS) scores preoperatively, on postoperative day 7, at 3 months postoperatively and at one year postoperatively. Reduction of VAS scores were recorded by preoperative minus postoperative VAS scores. Functional outcomes were evaluated by independent reviewers via administration of the Oswestry Disability Index (ODI) preoperatively, at 3 months postoperatively, and after one year of follow-up. Additionally, radiography was performed postoperatively, at 3 months postoperatively, at 6 months postoperatively, and every year thereafter in follow-up. Postoperative complications and neurologic deficits were also reviewed during follow-up.

Surgical outcome was assessed by a modified version of Odom’s criteria, which evaluates improvement or deterioration after surgery. In Odom’s criteria, “excellent” means complete recovery and return to previous activity, “good” means occasional back or leg pain and return to previous activity, “fair” means partial recovery and modified activities, and “poor” means no relief of the original symptoms or a worsening of symptoms. A result is considered “satisfactory” if the patient had “excellent” or “good’ in Odom’s criteria. The result was evaluated by independent research assistants.

### Statistical Analysis

SPSS software (version 13.0; IBM Corp., Armonk, NY, USA) was used to analyze the collected data. The two-tailed Student’s *t*-test was used for the continuous variables, and a Fisher’s exact test was used for group comparison of categorical variables. A *p*-value < 0.05 was set as the level of significance. A multiple regression analysis was also used in the evaluation of the data. Numerical data were presented as mean ± standard deviation, while categorical data were expressed in absolute frequencies.

## 3. Results

There were 49 patients (36 males and 13 females) included in this study. The overall mean age was 55.4 years (range 18–86 years), and the mean follow-up time was 47.3 months (range 15–150 months). Thirty-three patients underwent decompression-only surgery, and 16 patients underwent fusion surgery. The patient demographic data are summarized in Table 1. In the decompression group, there were 24 male and 9 female patients with a mean age of 57 ± 15.2 years. In the fusion group, there were 12 male and 4 female patients with a mean age of 51.3 years. There were five L1-2 cases of HIVD (15.2%) in the decompression group and five L1-2 cases of HIVD (31.3%) in the fusion group. The preoperative mean VAS score was 6.8 in the decompression group and 7.2 in the fusion group. The preoperative mean ODI was 72.5 in the decompression group and 70 in the fusion group. The perioperative data and follow-up data by a multiple regression analysis are summarized in Table 2.

The VAS scores showed improvement of both groups postoperatively and the improvement was not significantly different between the two groups. The mean reduction of VAS score (improvement between preoperative VAS and the VAS at one-year follow-up) was 4.8 in the decompression group and 5.4 in the fusion group (*p* = 0.450). During the follow-up, 3-month postoperative ODI showed better results in the fusion group (37 ± 17.5 in the decompression group versus 27.5 ± 5.5 in the fusion group, *p* = 0.09 in multivariate regression analysis). However, there were no significant differences in 12-month postoperative ODI and the mean reduction of ODI between the two groups. Moreover, 10 patients (76.9%) in the decompression group and 5 patients (83.3%) in the fusion group reported an improvement of their weakness. The final “satisfactory” rate was 75.5% for all patients, 66.7% in the decompression group, and 93.8% in the fusion group (*p* = 0.034).

Three patients in the decompression group suffered a recurrence of HIVD. Not only did clinical symptoms return during follow-up but the diagnosis was confirmed by magnetic resonance imaging. One of the patients underwent a nerve block procedure twice to control radiculopathy. The other two patients had failed conservative treatment and received wide laminectomy and fusion procedures that could provide adequate nerve decompression and gain immediate spinal stability (Figure 1). Another four patients (3 in the decompression group and 1 in the fusion group) complained about leg pain postoperatively that subsided after a nerve block procedure. Wound discharge problems were noticed during follow-up in two patients from each group. They received superficial debridement surgery under the diagnosis of stitch abscess without evidence of further infection. In this series, there was no mortality and no major complications, such as postoperative neurologic deficit or deep wound infections. There was no evidence of adjacent segmental disease (ASD) found in the fusion group for at least 1.5 years of follow-up.

## 4. Discussion

The surgical goal in upper lumbar HIVD is to release the compromised dura and nerve roots, as is the surgical goal in lower lumbar HIVD. The unique anatomic structure of the upper lumbar region, however, such as the narrower spinal canal and the level of the conusmedullaris, might lead to difficulty and greater risk for surgical intervention. In this present study, 33 patients (67.4%) received decompression-only surgery with a 66.7% “satisfactory” rate. Three patients in the decompression group had poor outcomes due to a recurrence of HIVD and needed revision fusion surgery or repeated nerve blocks. Eight patients had a “fair” outcome because of some accompanying degrees of persistent back pain, leg pain, or persistent motor weakness postoperatively. On the one hand, limited decompression was performed to avoid iatrogenic instability. On the other hand, residual stenosis or retained disc fragment due to inadequate decompression might cause the nerve structure to be compromised even after surgery.

Wide laminectomy was necessary in some patients to expose the disc space but led to facet destruction and spinal instability due to the narrow distance between the two pars interarticularis at the upper lumbar levels [10]. Additionally, preoperative discogenic problems without stabilizers and postoperative mechanical back pain after extended decompression procedures might also cause postoperative pain with unsatisfactory results. In fact, the methods of decompression might need to be individualized for patients with upper lumbar HIVD according to their clinical characteristics and presentations on imaging which may lead to different surgical approaches. During preoperative evaluation, a fusion surgery may be considered if patients have a huge central herniated disc found from image study. Iatrogenic instability might develop after extended decompression. Intraoperatively, once laminectomy of more than 50% of the lamina or facetectomy has been performed, fusion surgery should be considered to rebuild spinal stability.

In this present series, 16 patients (32.6%) underwent fusion surgery and had a greater “satisfactory” rate (93.8%) than with decompression alone (66.7%). The ODI scores at the three-month postoperative follow-up also showed better results in the fusion group than in the decompression group. Notably, no recurrence was reported in the fusion group. The reason for the difference between the two groups might be that the fusion surgery could achieve adequate decompression with immediate spinal stability, thus avoiding subsequent problems after decompression-only surgery and providing steady outcomes.

In an earlier study, the surgical treatment of upper lumbar HIVD reported poorer outcomes than in the surgical treatment of lower lumbar HIVD [2,3,7,11]. One important reason is the unique anatomic limitations of the surgical approach. Adequate decompression and spinal stability may be difficult to achieve for spinal surgeons during operations. Another reason might be a delay in the correct diagnosis due to unclear clinical symptoms and the low incidence rate. Prolonged nerve compression can cause chronic inflammation, tissue fibrosis, and irreversible damage. In our current practice, fusion surgery after routine discectomy is not recommended according to the updated guidelines [9]. For some indications, including evidence of spinal instability, severe degenerative change, chronic low back pain, or if the patient participates in heavy labor, fusion surgery may be considered. However, the characteristics in the upper lumbar levels may result in unfavorable surgical outcomes in treating HIVD than in the lower lumbar levels using decompression-only surgery. Fusion surgery may be a more secure method for not only providing adequate nerve decompression but also for immediate stability.

Several surgical approaches have been reported for treating patients with upper lumbar HIVD [12,13,14,15,16,17,18,19]. The choice of approach should be individualized according to the clinical symptoms, signs, and imaging findings. If there is bilateral radiculopathy, wide laminectomy or bilateral laminotomy might be performed. The nerve structure should be adequately relieved during the operation. Imaging findings, including herniated disc size, type, location, whether or not there has been migration, and degree of spinal cord compression, should be evaluated carefully. Kim et al. [4] presented a 41-patientseries with an 80.5% significant improvement after discectomy. In that study, discectomy was performed in three ways: Unilateral laminectomy for the majority of patients, bilateral laminectomy, and a transdural approach. In comparison to our study, an 80.5% satisfactory rate is better than the 66.7% in our decompression-only group. However, there were still 20% of patients with no improvement or even worsening symptoms after decompression surgery. In addition to open surgery, Wang et al. [14] presented a series of HIVD in the thoracolumbar junction (T12 to L3 level) treated by invasive transforaminalinterbody fusion surgery. In the study, fusion surgery was considered as a safe and effective procedure to improve the clinical and radiographic outcomes during follow-up. In addition to open surgery, endoscopic lumbar discectomy has become an alternative treatment. Ahn et al. [15] showed that 80% of a 45-patientseries diagnosed with upper lumbar HIVD reported a satisfactory outcome after undergoing percutaneous endoscopic lumbar discectomy (PELD) at L1-2 and L2-3 levels. Xinet al. [16] reported a modified translaminar osseous channel-assisted PELD to treat migrated and sequestrated discs of the upper lumbar region. The patients had pain relief immediately after surgery and good functional scores at one-year follow-up. In addition, Pan et al. [20] compared PELD to traditional open discectomy (OD) in a randomized controlled trial. During the follow-up, the clinical outcomes were better in the PELD group, and the levels of C-reactive protein, creatine phosphokinase, and interleukin 6 were all lower in the PELD group than in the OD group. Another study showed similar results [21]; however, opposite results have also been reported [22]. PELD was considered less invasive to soft tissue and with a faster recovery. Nevertheless, this was a technique-dependent surgery with learning curves and patient selection also playing important roles. A case report [18] showed a successful result of microendoscopy-assisted lumbar discectomy via the transforaminal approach to treat recurrent upper lumbar disc herniation. As minimally invasive surgery (MIS) became popular, the transforaminal approach via endoscopy became an alternative to the conventional posterior approach. Nevertheless, a study with larger patient numbers and a longer follow-up period is urgently needed to demonstrate the advantages of MIS. Fusion surgery may be considered whenever extended decompression is necessary and spinal stability is a concern.

There are some limitations in the present study. First, the nature of a retrospective design may have introduced some bias. Second, the small patient numbers in both groups causes difficulty for rigorous data analysis. Nevertheless, due to a very low incidence of upper lumbar HIVD, this study had a relatively large number of patients for a discussion of optimal surgical approaches and outcomes. In addition, ASD, which generally develops during years postoperatively, is the most considerable complication after fusion surgery and a longer follow up is necessary to evaluate potential problems arising out of ASD. Although the study design was not a prospective randomized trial, the two groups discussed here appear comparable regarding influences on clinical outcomes for treating upper lumbar HIVD.

## 5. Conclusions

The overall surgical outcomes were acceptable in this present study without major complications during a relatively short period of follow-up. The choice of approach should be individualized by clinical characteristics and imaging findings. Fusion surgery should be considered whenever extended laminectomy is needed and iatrogenic instability is of concern. More reliable “satisfactory” rates and better functional scores at the three-month postoperative follow-up were reported in the fusion group.

## Figures and Tables

**Figure 1 jcm-08-01435-f001:**
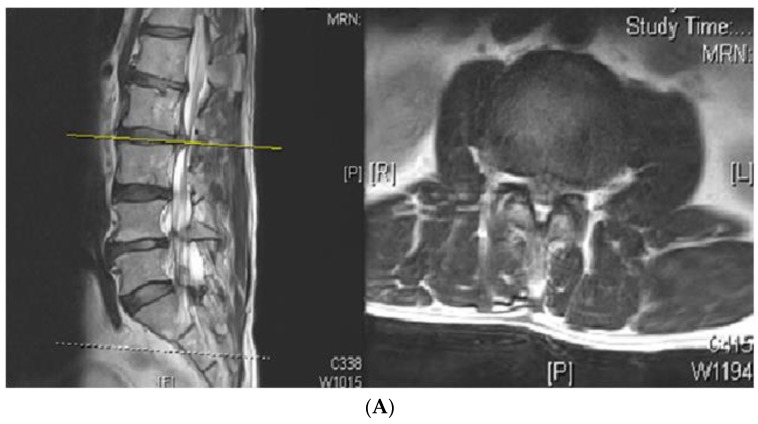
A 51-year-old male patient suffered from right radiculopathy due to a L2-3 herniated intervertebral disc (HIVD), on the right side (**A**), and received decompression surgery. The symptoms were relieved after operation. However, four months later, the symptoms developed again, and magnetic resonance imaging (MRI) showed recurrence of HIVD at L2-3 (**B**). Revision surgery was then performed with instrumentation plus cage (**C**, right). During follow-up for 4 years, the radiography showed stable implants (**C**, left).

**Table 1 jcm-08-01435-t001:** Patient demographics and characteristics.

Variables	Decompression Group (D) (*n* = 33)	Fusion Group (F) (*n* = 16)	*p*-Values
Age (years)	57 ± 15.2	51.3 ± 12.4	0.201
Sex (male/female)	24/9	12/4	0.267
Level (*n*)	L1-2 (5) L2-3 (28)	L1-2 (5) L2-3 (10) Both (1)	0.189
Follow-up (months)	51.2 ± 33	37.4 ± 19.6	0.138
PreOP VAS	6.8 ± 1.7	7.2 ± 1.1	0.487
PreOP ODI	72.5 ± 5.8	70 ± 7	0.196
Motor weakness	13 (39.4%)	6 (37.5%)	0.243
Improvement of motor weakness	10/13 (76.9%)	5/6 (83.3%)	0.443
Satisfactory rate	22 (66.7%)	15 (93.8%)	0.034 *

L1-2: HIVD at the L1-2 level; L2-3: HIVD at L2-3 level;Both: HIVD at L1-2 and L2-3 levels; PreOP VAS: Preoperative visual analog scale (VAS) score; PreOP ODI: Preoperative Oswestry Disability Index (ODI) score. * Significant difference between two groups; *p* < 0.05.

**Table 2 jcm-08-01435-t002:** Perioperative data and outcomes.

Variables	Univariate Regression Analysis			Multivariate Regression Analysis ^†^
	B(95% CI)	*p*-Value	B(95% CI)	*p*-Value
Blood loss	23.9 (−15.78, 63.58)	0.232	26.52 (−14.48, 67.51)	0.199
OP time	25.17 (−1.92, 52.26)	0.068	27.65 (0.356, 54.94)	0.047 *
VAS				
PreOP	0.34 (−0.64, 1.31)	0.487	0.51 (−0.47, 1.49)	0.3
7 days	0.3 (−0.73, 1.12)	0.561	0.27 (−0.74, 1.28)	0.595
6 month	−0.29 (−1.02, 0.45)	0.434	−0.43 (−1.17, 0.3)	0.244
12 month	−0.25 (−1.13, 0.63)	0.572	−0.37 (−1.24, 0.49)	0.387
ODI				
PreOP	−2.55 (−6.45, 1.36)	0.196	−2.78 (−6.67, 1.1)	0.156
3 month	−9.47 (−18.67, −0.27)	0.044 *	−11.85 (−20.52, −3.18)	0.009 *
12 month	−1.894 (−8.05, 4.26)	0.539	−3.93 (−9.26, 1.4)	0.144

^†^ Adjusted for age, sex; PreOP VAS: Preoperative visual analog scale (VAS) score; PreOP ODI: Preoperative Oswestry Disability Index (ODI) score; OP: Operative time; * Significant difference between two groups; *p* < 0.05.

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
