# Peer review of "Surgical Outcomes for Upper Lumbar Disc Herniation: Decompression Alone versus Fusion Surgery"

_jcm, 2019, doi:10.3390/jcm8091435_

Round 1
Reviewer 1 Report
Comments for Dr Tung-Yi Lina’s article,
The Authors statistically proved the favorable surgical outcomes of fusion surgery rather than decompression alone, for upper lumbar herniated intervertebral disc (HIVD), defined as L1–2 and L2–3 levels.
This is an interesting article regarding the treatment for upper HIVD. Before resubmitting the article, we request some major and minor changes as follows.
Major point:
In this article, 49 patients were included: 31 patients underwent 20 decompression-only surgery and 16 patients underwent fusion surgery. The one of critical points of this article is absence of consideration to other treatment opportunity. For example, transforaminal approach of full-endoscopic spinal surgery may possible to preserved facet joint of upper HIVD. The authors cited the article of Wu et al.[15], although this article seems to be inappropriate regarding on the poor operative outcome. Please consider to cite other articles as follows;
Ahn Y, Lee SH, Lee JH, Kim JU, Liu WC. Acta Neurochir (Wien). 2009 Mar;151(3):199-206. (cited as [18])
Xin Z, Liao W, Ao J, Qin J, Chen F, Ye Z, Cai Y. Biomed Res Int. 2017;2017:3069575.
The authors reported that 3-month postoperative Oswestry 23 Disability Index (ODI) was significantly better in the fusion group. Although 3-month after operation is too short for evaluate the significance of fusion procedure. And the authors described that there was no mortality and no major complications, such as postoperative neurologic deficit or deep wound infections However the most considerable complication of fusion surgery is the adjacent segmental disease (ASD), and the complication generally occur several years after fusion operation. If possible, please mention about ASD at least 2 year after fusion operation. Alternatively, the authors should clarify and rewrite the outcome is the short period of that. Please indicate the differences from previous similar studies and the results.
Int J Surg. 2014;12(5):534-7. This study shows the changes of interleukin-6 (IL-6), C-reactive protein (CRP) and creatine phosphokinase (CPK)
Med Sci Monit. 2016 Feb 18;22:530-9. This study also shows the changes of IL-6 CRP, and CPK.
Similar to above comment, some investigator did not find significant differences in the levels of serum inflammatory factors (TNF-α and CRP and oxidative stress indicators MDA, MPO, SOD and TAC). Please also discuss against such opposite opinion.
Exp Ther Med. 2018 Jan;15(1):295-299.
Minor points;
Please correct minor misspelling (ex. Line 203 As minimall yinvasivesurgery (MIS)→minimally invasive surgery)
Author Response
Thank you for your comments concerning our manuscript. We are pleased to have an opportunity to revise our paper.
We have carefully considered the valuable comments and have made corrections and revisions which we hope will meet with your approval for publication.
The detailed changes in the revised manuscript are underlined (highlighted in the tables), and the responses to each reviewer's questions are listed point-by-point:
Reviewer 1
Major point:
Response:
In our article, alternative treatments were also considered and some references were cited in the Discussion section, paragraph 5. We removed the inappropriate citation and cited new references in our article from the Discussion section, 5th paragraph, from line 215. During the follow-up period, 3-month ODI was significantly better in the fusion group than in the decompression group. Indeed, it was a short-term outcome. We revised the article and mentioned that “There was no evidence of adjacent segmental disease (ASD) found in the fusion group for at least 1.5 years of follow- up.” in the Results section, 3rd paragraph, line 145-146. Indeed, adjacent segmental disease may develop several years later operation, just as the reviewer stated. Therefore, we also revised the limitations section accordingly, in the Discussion, from line 241. In addition, the conclusion was revised to include that the outcome was based on a relatively short postoperative follow-up period ( in Conclusion section, line 251). We also revised the manuscript in Discussion section, paragraph 5, from line 221 and cite these studies as references. The opposite opinion was also mentioned.
Minor points:
Response:
We corrected the misspelled words. The language of the manuscript was also rechecked.
Reviewer 2 Report
The authors are dealing with an interesting topic, although this, the manuscript has several drawbacks among which the most important are:
The manuscript is poorly written due mainly to syntax errors
Were there any differences regarding imaging findings and/or clinical picture among the two groups or the decision for fusion was taken intra-operatively?
Among the 16 patients with fusion were included also the patients suffered from recurrent disc herniation and if yes what was the period between surgery and recurrence?
What do the authors mean by "the final satisfactory rate" and how do they measure it.
A multiple regression analysis would be more appropriate in the elaboration of the statistical data

Author Response
Thank you for your comments concerning our manuscript. We are pleased to have an opportunity to revise our paper.
We have carefully considered the valuable comments and have made corrections and revisions which we hope will meet with your approval for publication.
The detailed changes in the revised manuscript are underlined (highlighted in the tables), and the responses to each reviewer's questions are listed point-by-point:
Reviewer 2
Response:
The paper has been carefully revised by a professional language editing service to improve the grammar and readability. An editing certificate can be provided. Preoperatively, a fusion surgery may be considered if a patient had a huge central herniated disc. Iatrogenic instability might develop after extended decompression. Intraoperatively, laminectomy of more than 50% of the lamina or facetectomy may result in instability and might require fusion surgery. We revised the Discussion section accordingly. (2nd paragraph, from line 170 to 174) There was no recurrent disc herniation in the fusion group. (line 178 to 179) The final satisfactory rate was assessed by a modified version of the Odom criteria. (In Materials and Methods section, 3rd paragraph, line 91 to 98) This method is not very accurate but is easy to apply and used worldwide:The Odom criteria, established in 1958, are a widely used, 4-point rating scale for assessing the clinical outcome after cervical spine surgery. Later, it was also applied to lumbar discectomy. A result is considered satisfactory if the patient had an “excellent” or “good’ rating based on the Odom criteria.
Odom GL, Finney W, Woodhall B: Cervical disc lesions. JAMA 166:23-28, 1958
Indeed, a multiple regression analysis was more appropriate, and the analysis was performed. The 4th paragraph in Materials and Methods was revised and also Tables I and II were revised accordingly.
Round 2
Reviewer 1 Report
The authors responded carefully to each comment.
The responses make strongly secure the author’s opinion.
I am pleased to inform you that this article has been accepted for publishing on Journal of Clinical Medicine.